# The mitogenome of *Phytophthora agathidicida*: Evidence for a not so recent arrival of the "kauri killing" *Phytophthora* in New Zealand

Richard C. Winkworth[1,2]*, Stanley E. Bellgard[2], Patricia A. McLenachan[2], Peter J. Lockhart[1,2]

**1** Bio-Protection Research Centre, Massey University, Palmerston North, New Zealand, **2** School of Fundamental Sciences, Massey University, Palmerston North, New Zealand

* r.c.winkworth@massey.ac.nz

**Data Availability Statement:** The reported sequence data are available from GenBank using the accession numbers listed in Table 1 and are provided in the Supporting Information files.

## Abstract

*Phytophthora agathidicida* is associated with a root rot that threatens the long-term survival of the iconic New Zealand kauri. Although it is widely assumed that this pathogen arrived in New Zealand post-1945, this hypothesis has yet to be formally tested. Here we describe evolutionary analyses aimed at evaluating this and two alternative hypotheses. As a basis for our analyses, we assembled complete mitochondrial genome sequences from 16 accessions representing the geographic range of *P. agathidicida* as well as those of five other members of *Phytophthora* clade 5. All 21 mitogenome sequences were very similar, differing little in size with all sharing the same gene content and arrangement. We first examined the temporal origins of genetic diversity using a pair of calibration schemes. Both resulted in similar age estimates; specifically, a mean age of 303.0–304.4 years and 95% HPDs of 206.9–414.6 years for the most recent common ancestor of the included isolates. We then used phylogenetic tree building and network analyses to investigate the geographic distribution of the genetic diversity. Four geographically distinct genetic groups were recognised within *P. agathidicida*. Taken together the inferred age and geographic distribution of the sampled mitogenome diversity suggests that this pathogen diversified following arrival in New Zealand several hundred to several thousand years ago. This conclusion is consistent with the emergence of kauri dieback disease being a consequence of recent changes in the relationship between the pathogen, host, and environment rather than a post-1945 introduction of the causal pathogen into New Zealand.

## Introduction

The genus *Phytophthora* currently consists of ~180 formally described species and hybrids [1]. While this globally important group of pathogens affects a great many plant species, individual *Phytophthora* species differ markedly in host range; some have a limited number of hosts (e.g., *P. infestans* (Mont.) de Bary) while others have upwards of a hundred potential hosts (e.g., *P. cinnamomi* Rands, *P. nicotianae* Breda de Haan) [2]. Despite a growing understanding of

**Funding:** P.J.L. and R.C.W., BPRC_MU_2016_1, BioProtection Research Centre (https://bioprotection.org.nz). The funders had no role in study design, data collection and analysis, decision to publish, or preparation of the manuscript. R.C.W. and P.J.L., NP94607, Massey University (https://massey.ac.nz). The funders had no role in study design, data collection and analysis, decision to publish, or preparation of the manuscript. S.E.B, C09X1704, New Zealand Ministry of Business, Innovation and Employment Strategic Science Investment Fund (https://www.mbie.govt.nz/science-and-technology/science-andinnovation/funding-information-and-opportunities/investment-funds/strategic-science-investmentfund/). The funders had no role in study design, data collection and analysis, decision to publish, or preparation of the manuscript.

**Competing interests:** The authors have declared that no competing interests exist.

species diversity (e.g., [1, 3]) and evolutionary relationships (e.g., [4–6]), the origins of many *Phytophthora* species remain uncertain (e.g., [3, 7]).

The genus *Agathis* (Araucariaceae) consists of ~13 species of tropical to warm temperate gymnosperms distributed across Malesia, Australia and New Zealand. The single New Zealand species, kauri (*Agathis australis* (D.Don) Loudon), is endemic and the dominant tree in lowland forests of the northern North Island [8]. A large, long-lived species kauri is an ecosystem engineer influencing the composition of associated communities via affects on soil chemistry [9–12]. The species is culturally significant to indigenous Māori [13] but over the last 175 years kauri forests have been reduced to less than 10% of their pre-European extent by commercial felling, clearance for farming, and natural fires [8, 14, 15]. The extreme reduction and fragmentation of kauri forests makes this species highly vulnerable to other threats.

Gadgil [16] was the first to report a root rot of kauri. His pioneering work associated disease symptoms with a *Phytophthora* initially identified as *P. heveae* A.W. Thomps. Assumed to be restricted to an offshore island, it was not until symptoms were found on the mainland in 2006 that this disease was recognized as a wider threat [9, 13, 17]. Subsequent work has found that although the "kauri killing" *Phytophthora* and *P. heveae* both belong to *Phytophthora* clade 5 [18], the two are morphologically and genetically distinct [19]. The species associated with socalled kauri dieback is now formally described as *P. agathidicida* B.S. Weir, Beever, Pennycook & Bellgard [19]. Of the six *Phytophthora* species reported from kauri forest soils, *P. agathidicida* is one of only two previously linked to disease in the field [9, 13, 20]. Studies to date indicate that *P. agathidicida* is highly pathogenic toward kauri. For example, Beever *et al.* [9] identified symptoms of *P. agathidicida* infection in most size classes of kauri and Horner & Hough [17] observed 100% mortality within four months of inoculation for two-year old kauri seedlings in a control group. The other *Phytophthora* recognized as disease causing in kauri, *P. cinnamomi*, has yet to be directly linked to the now widespread dieback [9].

*Phytophthora* clade 5 currently consists of five taxa–*P. agathidicida*, *P. castaneae* Katsura & K. Uchida, *P. cocois* B.S. Weir, Beever, Pennycook, Bellgard & J.Y. Uchida, *P. heveae*, and *P.* sp. "novaeguineae". The geographical distributions of the currently recognized species and their plant hosts imply an eastern Asian and Pacific centre of diversity [19]. However, as is common for members of *Phytophthora* the origins of the individual taxa remain uncertain (e.g., [7]). In the case of kauri dieback, the history of host and pathogen appear to be in striking contrast. *Agathis australis* has been long isolated in New Zealand and is genetically distinct from other members of the Araucariaceae [21–23]. In contrast *P. agathidicida* is genetically very similar to the other members of clade 5 suggesting that this species has a much more recent origin [19]. Although analyses to date are consistent with *P. agathidicida* having only recently established in New Zealand, when and from where this pathogen originated is yet to be established [19].

Biogeographic studies often use divergence time estimates from molecular data to distinguish between historical scenarios (e.g., [24–26]). Depending on the question and the available sampling such studies may use either stem or crown age estimates; that is, estimates for the age of the most recent common ancestor (MRCA) of the focal taxon and its sister group or the MRCA of extant diversity within the focal taxon, respectively. In the case of *P. agathidicida*, species diversity in *Phytophthora* clade 5 is poorly characterised and as a result stem age estimates for *P. agathidicida* could be biased by the failure to include, as yet, undescribed taxa. If so then we are likely to draw the erroneous conclusion that this pathogen has a long history in New Zealand. The alternative is to use age estimates for the extant diversity of *P. agathidicida*. In this case, if the estimated crown age were more recent than the hypothesised arrival time then we would infer that the pathogen diversified following arrival in New Zealand. In contrast, if the crown age were older than the hypothesised arrival time we would need to consider

two possibilities. We could either reject the hypothesis in favour of an alternative arrival time or infer that several genetically distinct strains had been introduced New Zealand.

In the current study we considered three scenarios for the arrival of *P. agathidicida* in New Zealand. The first is a recent incursion. A common suggestion has been that *P. agathidicida* was introduced into New Zealand on imported forestry equipment or planting materials during the late 1940s (see [27]). Assuming a recent incursion we would predict a crown age for *P. agathidicida* of less than 70 years old. A second possibility is that pathogen arrival was associated with large scale European settlement of New Zealand between 1845 and the 1940s. During this period kauri forests were heavily disturbed and many of the organisms now considered pests in New Zealand were introduced [28]. Assuming this scenario we would predict a crown age for *P. agathidicida* of 70–175 years old. The third possibility is that *P. agathidicida* was present in New Zealand prior to 1845. In this case, the pathogen may have evolved in New Zealand, arrived via natural dispersal or as the result of earlier human movements (e.g., pre-1845 European visitors). Assuming this scenario we would predict the crown of *P. agathidicida* to be more than 175 years old.

In this study we have assembled complete mitochondrial genomes for 21 *Phytophthora* clade 5 isolates; 16 accessions of *P. agathidicida* from across the geographic distribution of this pathogen and five representing the other *Phytophthora* clade 5 taxa. From these we infer phylogenetic relationships and estimate divergence times for *P. agathidicida*. We discuss our results in terms of pathogen history as well as understanding the spread and management of kauri dieback disease.

## Materials and methods

### Sampling, sample preparation and genome sequencing

We obtained four accessions of *P. agathidicida* and five representing other *Phytophthora* clade 5 taxa from the International Collection of Microorganisms from Plants (ICMP) culture collection (Table 1). The selected accessions were cultured on *Phytophthora*-selective media [29] in the dark at 18˚C for 7–10 days. Agar plugs were excised from each plate and incubated overnight at 56˚C with 180 μL ATL buffer (Qiagen, Hilden, Germany) and 20 μL proteinase K (20 mg/ml; Qiagen). The tubes were then briefly centrifuged before genomic DNA was extracted from the supernatant using the QIAcube® instrument and the QIAamp® DNA mini QIAcube kit (Qiagen) according to manufacturer's protocol. Shotgun sequencing libraries were prepared from DNA extracts by the Massey Genome Service (Palmerston North, New Zealand) using Illumina Nextera DNA library preparation kits (Illumina, Inc., San Diego, CA). The Massey Genome Service also performed 2 × 250 base pair paired-end DNA sequencing on Illumina MiSeq instruments. Following sequencing reads were quality assessed using a combination of SolexaQA [30] and FastQC [31].

To expand the geographical range of our sampling we obtained SRA libraries for twelve accessions of *P. agathidicida* from Genbank (Table 1). Together the 16 included accessions broadly represent the geographic range of *P. agathidicida* (Fig 1).

### Mitochondrial genome assembly and annotation

For each accession we assembled complete mitochondrial genome sequences from quality-assessed paired-end Illumina reads using a combination of *de novo* and reference-based approaches. Preliminary *de novo* assemblies were made using idba_ud [32] with the resulting contigs compared to a collection of publicly available and unpublished Oomycete mitochondrial genome sequences (S1 Table) using BLAST [33]. Contigs with high similarity to the reference set were assembled into draft mitochondrial genome sequences using Geneious R9 [34].

**Table 1. Details of the sampled *Phytophthora* clade 5 accessions and corresponding mitochondrial genome sequences.**

| Accession no.[a] | Collection location[b] | Collection year | Mitochondrial genome sequence | | |
|---|---|---|---|---|---|
| | | | GenBank accession no. | Length (bp) | Average read coverage |
| *Phytophthora agathidicida* | | | | | |
| ICMP 16471 | Great Barrier Island | 1972 | MN883601 | 36826 | 2492.0 |
| ICMP 18244 | Pakiri, Auckland | 2008 | MT032126 | 36844 | 4458.4 |
| ICMP 18410 | Trounson Park, Northland | 2010 | MW287988 | 36826 | 4494.5 |
| ICMP 20275 | Whangapoua Forest, Coromandel | 2014 | MW287989 | 36835 | 493.5 |
| NZFS 3118[c] | Huia, Auckland | 2009 | BK014402 | 36839 | 15163.3 |
| NZFS 3126[c] | Piha, Auckland | 2006 | BK014403 | 36837 | 9505.9 |
| NZFS 3128[c] | Huia, Auckland | 2009 | BK014416 | 36841 | 3987.9 |
| NZFS 3616[c] | Great Barrier Island | 2001 | BK014404 | 36860 | 5711.5 |
| NZFS 3687[c] | Waipoua Forest, Northland | 2011 | BK014405 | 36846 | 2733.6 |
| NZFS 3770[c] | Great Barrier Island | 2006 | BK014406 | 36826 | 3502.5 |
| NZFS 3772[c] | Huia, Auckland | 2013 | BK011977 | 36824 | 3650.5 |
| NZFS 3815[c] | Coromandel | 2014 | BK014407 | 36851 | 19059.9 |
| NZFS 3869[c] | Arapohue, Northland | 2014 | BK014408 | 36870 | 13730.5 |
| NZFS 3885[c] | Ruawai, Northland | 2014 | BK014409 | 36823 | 7686.4 |
| NZFS 4289[c] | Raetea, Northland | 2010 | BK014410 | 36847 | 14708.0 |
| NZFS 4290[c] | Waipoua Forest, Northland | 2010 | BK014411 | 36846 | 18585.9 |
| *Phytophthora castaneae* | | | | | |
| ICMP 19450 | Lenhuachih, Taiwan | 1988 | MN883602 | 37083 | 3535.3 |
| *Phytophthora cocois* | | | | | |
| ICMP 16949 | Kauai, Hawaii | 1990 | MN883603 | 37078 | 1904.2 |
| ICMP 19685 | Port-Bouët, Cote D'Ivoire | 1991 | MT032127 | 37125 | 6857.7 |
| *Phytophthora heveae* | | | | | |
| ICMP 19451 | Selangor, Malaysia | 1927 | MN883604 | 37150 | 4165.4 |
| *Phytophthora* sp. "novaeguineae" | | | | | |
| ICMP 19637 | Papua New Guinea | — | MN883605 | 37072 | 5056.0 |

[a] Sampled accessions are from the International Collection of Microorganisms from Plants (ICMP) or New Zealand Forest Research Institute (NZFS) culture collections.

[b] Unless otherwise indicated collection locations are within New Zealand.

[c] Data was retrieved from the NCBI Sequence Read Archive, BioProject numbers SRX1116282, SRX1116283, SRX4575874-SRX4575886.

Genome drafts were then checked for assembly errors by mapping the quality-assessed reads directly to drafts using BWA [35]. Drafts were revised as appropriate following a visual inspection of mapped reads in Tablet [36].

Mitochondrial genome sequences were annotated on the basis of similarity to publicly available Oomycete genomes using a combination of Geneious and DOGMA [37].

## Phylogenetic and network analyses

A multiple sequence alignment for the complete mitochondrial genomes of all 21 accessions was constructed for phylogenetic analysis. An initial alignment was generated using ClustalO [38] and edited in Mesquite version 3 [39] to remove sections where less than 50% of the sequences were represented or where overlapping indels made alignment ambiguous.

Two methods were used to evaluate evolutionary relationships. First, we used RAxML v8.3.17 [40] together with a GTRGAMMA model to construct a maximum likelihood tree from the full matrix. Support for recovered relationships was evaluated using 1000 bootstrap

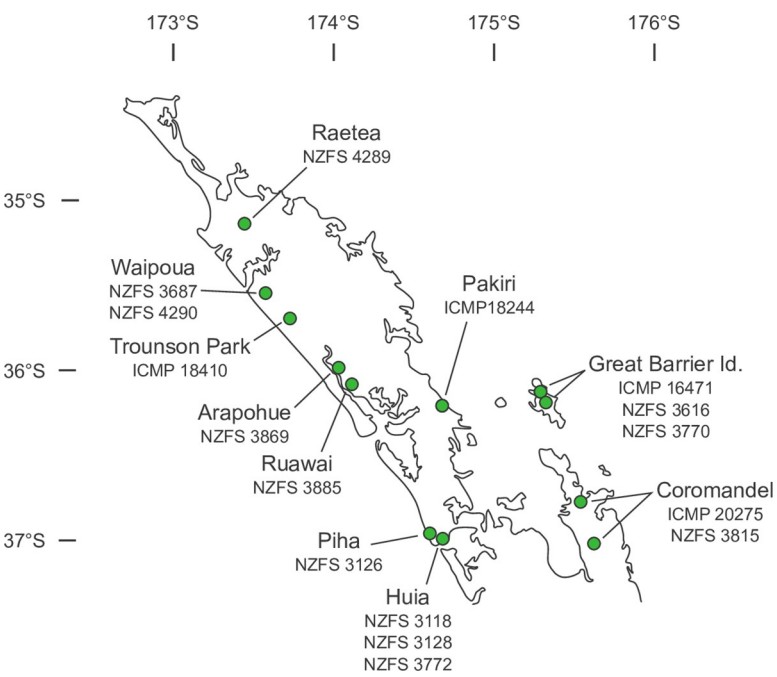

**Fig 1. Map of the northern North Island of New Zealand showing collection locations for sampled *Phytophthora agathidicida* accessions.** Base map used under a CC BY license, with permission from Free Vector Maps, original copyright 2014.

replicates. Network approaches are often used to represent potentially conflicting phylogenetic information from sequence data (e.g., [41]) or tree sets (e.g., [42]). To further examine the underlying structure of the data we conducted a Neighbor-Net analysis [43], as implemented in SplitsTree4 [44], using only the 16 *P. agathidicida* accessions.

## Divergence time estimates

Estimating divergence times from DNA sequence data requires that the molecular clock be calibrated. Divergence time analyses are usually calibrated by assigning ages to nodes based on fossil evidence, by incorporating sampling time information, or by constraining the overall evolutionary rate. In this case fossils are not available and preliminary analyses suggested that collection dates alone would not be sufficient to calibrate the phylogeny. Instead to calibrate the present analyses we have used evolutionary rate estimates derived from a collection of *P. infestans* mitogenomes sampled over a period of ~150 years [26, 45, 46]. Relying upon rate estimates from a single species has limitations (e.g., [47]) but similar estimates are not yet available for other *Phytophthora* species and comparisons with the results of Yuan *et al.* [48] suggest that for the mitochondrial genomes of *P. agathidicida* and *P. infestans* microevolutionary processes are broadly similar.

We first conducted a sequential analysis in which clock rate estimates for *P. infestans* were used to calibrate a subsequent analysis of *P. agathidicida*. Following Martin *et al.* [26] we estimated clock rates for *P. infestans* using BEAST version 1.8.1 [49]. A multiple sequence alignment for the 69 *P. infestans* mitochondrial genomes analysed by Martin *et al.* [26] was generated using ClustalO [38]. We then established five data partitions (i.e., one each for first, second and third codon positions of protein coding genes with further partitions for RNA

gene and non-coding sequences) and determined best-fit models of sequence evolution for each partition using jModeltest 2.2 [50, 51]. Our BEAST analysis used the best-fit model of sequence evolution and a strict clock [52] with a uniform clock rate prior for each partition, a skyride tree prior [53], and was calibrated using the sample collection dates (see [26]). Posterior distributions of parameters were estimated using Markov chain Monte Carlo (MCMC) sampling; samples were drawn every $1.0 \times 10^3$ generations from a total of $5.0 \times 10^7$ generations with 10% discarded as burn-in. The analysis was repeated with visual comparisons of estimated sample sizes (ESS) and the posterior distributions of parameters to evaluate mixing and convergence.

The posterior distribution of the clock rate for each *P. infestans* partition was then used as a prior on the clock rate of the corresponding partition in a BEAST2 [54] analysis of the 16 *P. agathidicida* mitochondrial genomes. Again, we generated a multiple sequence alignment using ClustalO [38], established five data partitions, and determined the best fit model of evolution for each partition using jModeltest. For each partition we also specified lognormal clock rate priors based on the mean and 95% highest posterior density (HPD) of the clock rate for the corresponding *P. infestans* partition. We then used nested sampling [55], as implemented in the NS package [55] for BEAST2 [54], to evaluate the fit of four coalescent tree priors (i.e., coalescent with constant population size, coalescent with exponential population growth, Bayesian skyline, and extended Bayesian skyline) and two clock models (i.e., strict and relaxed lognormal). Log Bayes factors [56] for each combination of tree prior and clock model were then compared; differences of 1.1–3.0, 3.0–5.0, and >5 were considered positive, strong and overwhelming support, respectively [56]. The age of the MRCA for the sampled *P. agathidicida* isolates was then estimated using BEAST2 [54] as well as the best fit tree prior, clock model, and models of sequence evolution. Posterior distributions of parameters were estimated using Markov chain Monte Carlo (MCMC) sampling using the same chain length, sampling density and burn-in percentage as for the initial *P. infestans* analysis; the analysis was repeated to evaluate mixing and convergence.

Next, we performed a simultaneous analysis in which tree and substitution models for *P. agathidicida* and *P. infestans* were independent, but partition clock rates were estimated jointly. We first used the same nested sampling [55] strategy as above to evaluate the fit of tree priors and clock models to the *P. infestans* data set. Using the most appropriate combinations of tree prior, substitution model, and clock model for each species we again estimated the age of the MRCA of the sampled *P. agathidicida*. For this analysis uniform priors on clock rate were used for each of the five sequence partitions. Posterior distributions of parameters were estimated using Markov chain Monte Carlo (MCMC) sampling using the same chain length, sampling density and burn-in percentage as for the initial *P. infestans* analysis; the analysis was repeated to evaluate mixing and convergence.

## Tree model adequacy

We examined tree model adequacy using the TMA package [57] for BEAST2 [54]. This approach uses the chronogram from an empirical analysis as the basis for simulating data sets against which to evaluate the empirical result. If the models used in the empirical analysis are adequate then mean values of an appropriate test statistic should fall within the 95% HPD for the simulated data sets.

In the present study we evaluated the chronograms from sequential and simultaneous analyses for both *P. agathidicida* and *P. infestans*. In each case the TMA run consisted of 300 simulations with model adequacy evaluated on the basis of tree height, tree length, and root-to-tip heights.

## Results

### Mitochondrial genomes

We assembled complete, circularised mitochondrial genome sequences for 16 *P. agathidicida* accessions as well as accessions representing four other *Phytophthora* clade 5 taxa; two accessions of *P. cocois* and single accessions of *P. castaneae*, *P. heveae* and *P.* sp. "novaeguineae". Average coverage values for these assemblies were 493.5–19059.9 sequence reads (Table 1).

The reported mitochondrial genome sequences can be obtained from Genbank (see Table 1 for accession numbers) and from the S1 Data. All 21 *Phytophthora* clade 5 sequences were highly similar. For example, they differed by less than 327 nucleotides in length (Table 1) and shared the same set of 39 protein-coding genes, 25 transfer RNAs and two ribosomal RNAs; gene order was also conserved across these genomes (Fig 2). Given the overall similarity in

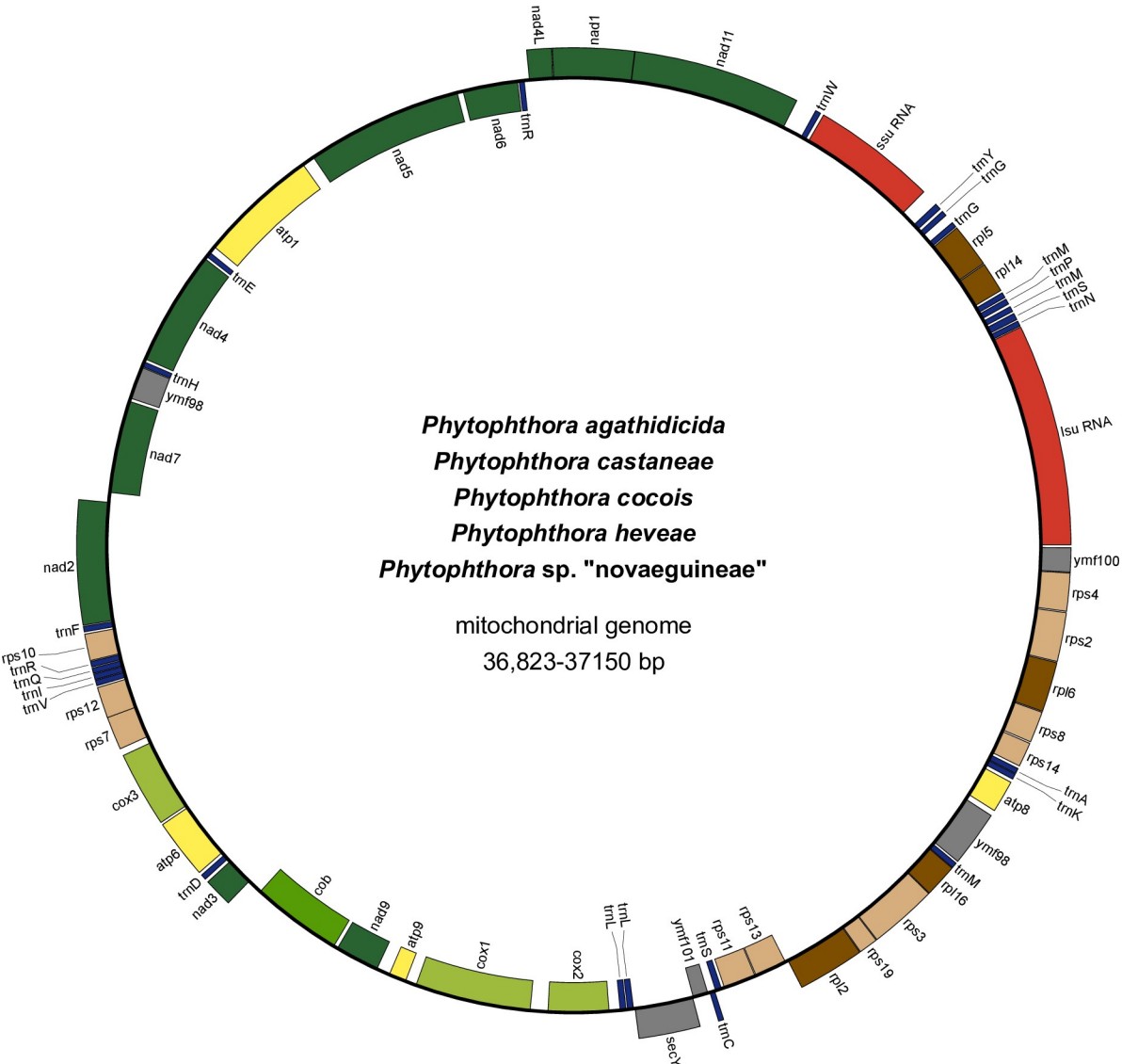

**Fig 2. The mitochondrial genomes of *Phytophthora agathidicida P. castaneae*, *P. cocois*, *P. heveae*, and *P.* sp. "novaeguineae".** Genes on the interior of the circle are transcribed in the forward direction; those on the exterior are transcribed in the reverse direction. The genome was drawn using OrganellarGenomeDraw [58].

length and gene content it is not surprising that GC content as well as the proportions of coding and non-coding DNA were also similar across all the genomes. Specifically, GC content ranged from 21.4% to 21.8% and the proportions of coding and non-coding DNA were 0.91–0.92 and 0.08–0.09, respectively.

The 16 *P. agathidicida* mitochondrial genome sequences differed in length by less than 47 nucleotides. This length variation was the result of 23 indels; of these, 19 were five nucleotides or shorter in length with the remainder 10 to 49 nucleotides long. All but four indels fell within non-coding sequences. The exceptions were all single nucleotide indels and were located within ribosomal genes. The *P. agathidicida* genome sequences also differed at the nucleotide sequence level. Our sample of 16 mitochondrial genomes contained 14 distinct sequence types. Of these 12 (75%) were recovered from single isolates with the remaining two sequences each recovered from a pair of accessions (i.e., ICMP 16471 and NZFS 3770, ICMP 20275 and NZFS 3815). Although the proportion of unique sequence types is high the sequences themselves exhibit little diversity at the nucleotide sequence level. Specifically, in pairwise comparisons the 14 sequence types differed from one another by 5–48 aligned sequence positions (i.e., 0.01–0.13%). The distribution of nucleotide sequence diversity was broadly consistent with the sizes of three functionally defined sequence partitions. The protein coding genes, which account for ~76.0% of the mitogenome sequence, contained 81.5% (53 of 65) of the variable sequence positions whereas the RNA genes (~16% of the sequence) and non-coding DNA (~8.0% of the sequence) each contained ~9.2% (6 of 65) of the variable positions. Although most variable positions fell within the protein coding partition, of the 39 protein coding genes 16 shared 100% identity with the remaining 23 having 1–5 variable positions.

## Phylogenetic and network analyses

The maximum likelihood analysis of the mitochondrial genome sequences for members of *Phytophthora* clade 5 resulted in a well-resolved and supported topology (Fig 3). In this tree the *P. agathidicida* accession (ICMP 16471) was sister to a clade containing the two *P. cocois* accessions; both these relationships received 100% bootstrap support. Among the three remaining accessions *P. heveae* and *P.* sp. "novaeguineae" form a clade, again with 100% bootstrap support.

Neighbor-Net analysis of the 16 *P. agathidicida* mitochondrial genome sequences suggests four geographically distinct groups (Fig 3). In the network three of the groups were clearly separated; one group consisted of accessions from Raetea (NZFS 4289) and Waipoua Forest (NZFS 3687, NZFS 4290), one included accessions from Great Barrier Island (ICMP 16471, NZFS 3616, NZFS 3770) and one the accessions from Arapohue (NZFS 3869), Coromandel (ICMP 20275, NZFS 3815), Ruawai (NZFS 3885), and Trounson Park (ICMP 18410). The remaining accessions–from Huia (NZFS 3118, NZFS 3128, NZFS 3772), Pakiri (ICMP 18244), and Piha (NZFS 3128)–were clustered within the network but did not clearly form a clade. Consistent with the relationships suggested by the Neighbor-Net analysis, mitogenome sequences from different geographic groups differed by 0.04–0.13% (14–48 aligned positions) whereas those from the same group differed by 0.00–0.08% (0–30 aligned positions). For example, accessions from Great Barrier Island (i.e., ICMP 16471, NZFS 3616 and NZFS 3770) from one another differed by 0.00–0.02% (0–8 aligned positions) but by 0.08–0.11% (29–41 aligned positions) from accessions from the Far North group (i.e., NZFS 3687, NZFS 4289 and NZFS 4290). In a few cases we included multiple accessions from the same local area; levels of nucleotide difference were similar to those within geographic groups. For example, the three accessions from Huia (i.e., NZFS 3118, NZFS 3128 and NZFS 3772) differed by 0.01–0.08% (5–30 aligned positions) whereas two of the three samples from Great Barrier Island (i.e., ICMP 16471 and NZFS 3770) shared 100% sequence identity.

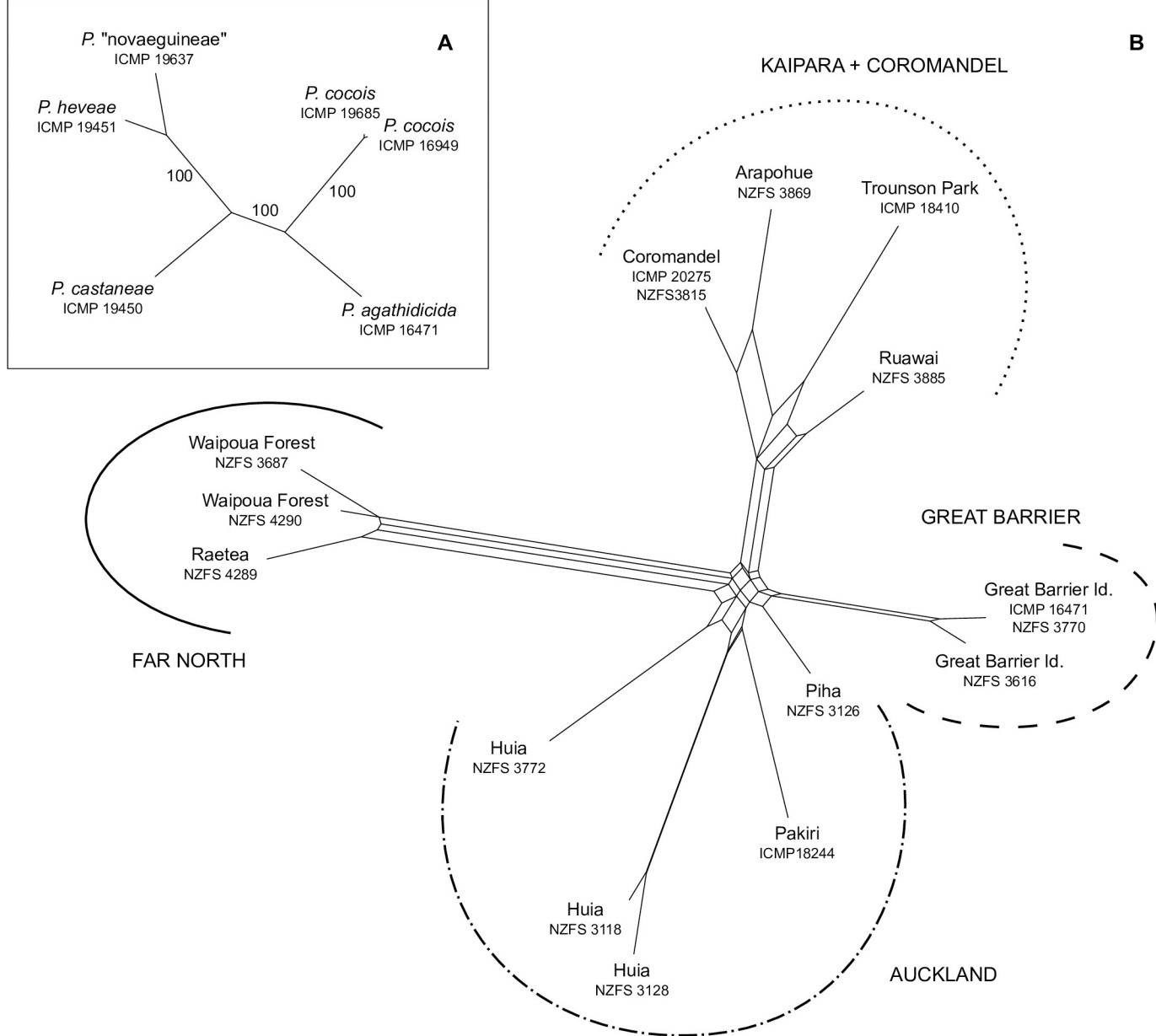

**Fig 3. Phylogenetic reconstructions for *Phytophthora agathidicida* and related *Phytophthora* clade 5 taxa. A.** Maximum likelihood tree describing relationships among members of *Phytophthora* clade 5. Tree obtained from analysis of complete mitochondrial genome sequences (likelihood = -54392.814), branches labelled with bootstrap values. **B.** Phylogenetic network describing relationships among the *Phytophthora agathidicida* accessions. Network obtained from Neighbor-Net analysis of complete mitochondrial genome sequences (Fit = 98.023). Terminals are labelled with collection location and accession number; four broad genetic groups are also labelled.

## Divergence time estimates for *P. agathidicida*

We first estimated clock rates for five mitochondrial sequence partitions in *P. infestans* using the approach of Martin *et al.* [26]. Our reanalysis of the Martin *et al.* [26] data suggested an age of 646.9 years (95% HPD, 409.7–932.7 years) for the MRCA of the sampled *P. infestans* accessions. Estimates of clock rates for the *P. infestans* mitochondrial partitions ranged from $3.58 \times 10^{-7}$ substitutions per site per year (95% HPD, $1.56 \times 10^{-7}$–$5.83 \times 10^{-7}$ substitutions per

site per year) for the RNA genes to $7.26 \times 10^{-6}$ substitutions per site per year (95% HPD, $4.32 \times 10^{-6}$–$1.03 \times 10^{-5}$ substitutions per site per year) for non-coding sites (S1 Fig).

For our sequential analysis we initially conducted nested sampling to identify the most appropriate clock rate models and coalescent tree priors for use with our *P. agathidicida* data set. Using best-fit substitution models and *P. infestans* clock rate estimates as priors, these analyses provided overwhelming support for the use of strict clocks (log Bayes factor = 38.89) and strong support of use of a coalescent tree prior with constant population size (log Bayes factor = 3.22) for the *P. agathidicida* data set. The subsequent divergence time analysis suggested an age of 304.4 years (95% HPD, 206.9–414.6 years) for the MRCA of all the sampled *P. agathidicida* accessions and of 66.8–231.5 years (95% HPD, 25.4–316.2 years) for the MRCA of each of the four geographical groups. In this topology accessions from the Great Barrier Island from a well-supported clade that appears to be nested within the Auckland grouping (Fig 4); however, consistent with the structure of our Neighbor-Net this arrangement was poorly supported. Estimates of clock rates for *P. agathidicida* ranged from $4.32 \times 10^{-7}$ substitutions per site per year (95% HPD, $2.39 \times 10^{-7}$–$6.51 \times 10^{-7}$ substitutions per site per year) for the RNA genes to $4.73 \times 10^{-6}$ substitutions per site per year (95% HPD, $4.65 \times 10^{-6}$–$6.38 \times 10^{-6}$ substitutions per site per year) for non-coding sites (S2 Fig).

For our simultaneous analysis we again used nested sampling to determine the most appropriate clock model and tree prior for the *P. infestans* data set. These analyses provided strong to overwhelming support for a Coalescent Bayesian Skyline tree prior (log Bayes factor = 3.00) and strict clock (log Bayes factor = 15.02) for the *P. infestans* data set. For *P. agathidicida* the simultaneous analysis recovered posterior distributions for the age of the MRCA sampled

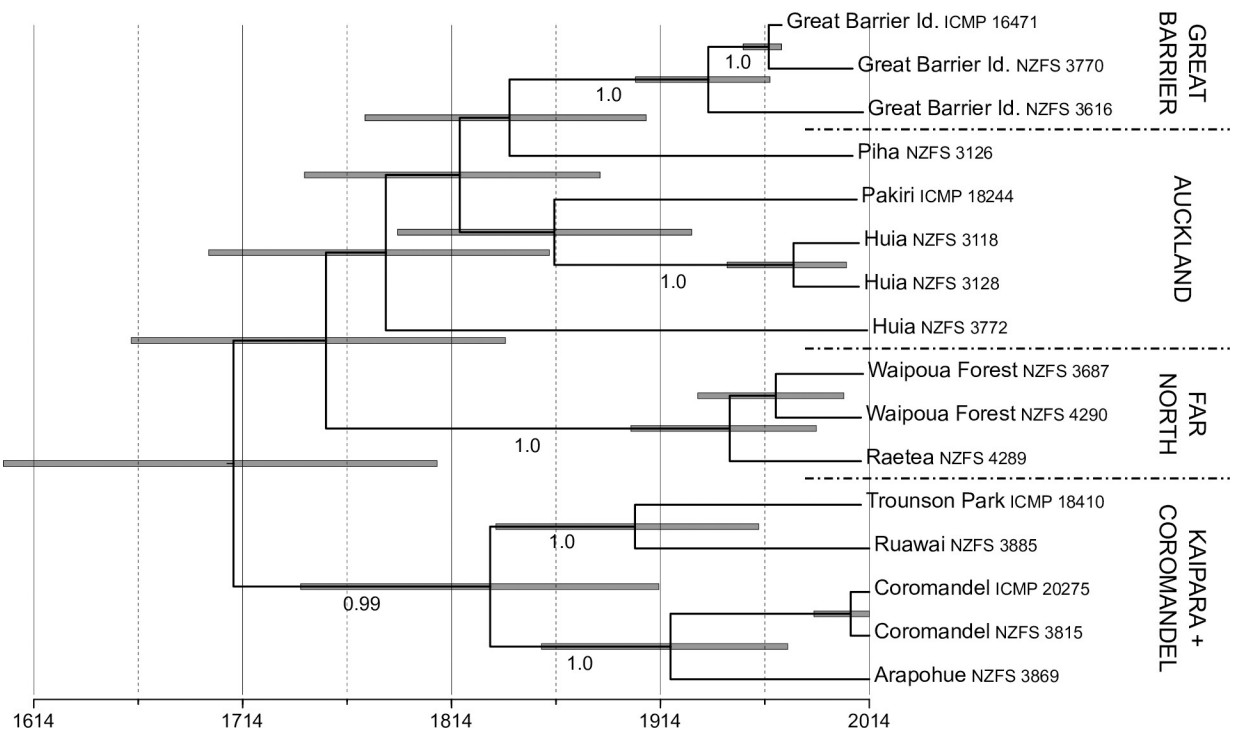

**Fig 4. Time-calibrated phylogeny from the sequential analysis of sampled *P. agathidicida* mitochondrial genomes.** Branch lengths are proportional to time with grey bars indicating the 95% highest probability densities. Terminals are labelled with collection location and accession number; four broad genetic groups are also labelled.

accessions and clock rates for sequence partitions that were highly similar to those of the sequential analyses. Specifically, the MRCA was estimated to be 303.0 years old (95% HPD, 210.1–405.2 years) with clock rates ranging from $3.53 \times 10^{-7}$–$5.73 \times 10^{-6}$ substitutions per site per year (95% HPD, $2.31 \times 10^{-7}$–$7.07 \times 10^{-6}$ substitutions per site per year) (S1 and S2 Figs). In this case the ages of the MRCA for each of the four geographical groups ranged from 66.2– 229.7 years (95% HPD, 24.8–314.6 years); again, the Great Barrier Island clade is nested within the Auckland grouping.

## Tree model adequacy

We used tree model adequacy to assess whether the tree models applied in our sequential and simultaneous analyses could have generated the observed data. For the parameters tree height, tree length, and root-to-tip heights we compared the mean of the posterior distribution to the 95% HPD for a collection of 300 simulated data sets. Without exception mean parameter values fell within the corresponding 95% HPD for the simulated data. For example, for tree height, which corresponds to the age of the MRCA, the observed mean value and 95% HPD for simulated data were 304.4 and 267.3–535.7 years, respectively. The models used in our analyses were therefore considered adequate.

# Discussion

## Diversity of clade 5 mitochondrial genomes

We assembled complete mitochondrial genomes from 21 accessions representing all currently recognised members of *Phytophthora* clade 5. These mitogenome sequences are highly similar; for example, they differed little in size sharing both gene content and arrangement. Mitogenome size and gene content is also broadly similar to those reported for other *Phytophthora* species. For example, all available *Phytophthora* mitogenomes contain the same 35 protein-coding genes of known function (i.e., 18 respiratory chain proteins, 16 ribosomal proteins, and an import protein) and two ribosomal RNA genes (e.g., [59, 60]). However, despite these similarities no previously reported *Phytophthora* mitogenome has the same structural arrangement as that identified for the clade 5 taxa.

Our sample of 16 *P. agathidicida* mitogenomes contained 14 distinct sequence types that in pairwise comparisons differed by 5–48 aligned sequence positions (i.e., 0.01–0.13%). Two previous studies [19, 61], both similar to the present study in the size and geographical range of sampling, have also reported estimates of sequence diversity for *P. agathidicida*. Most recently, Guo *et al.* [61] reported pairwise sequence differences for whole genome comparisons that are very similar to those for mitogenome sequences. Specifically, their sample of genomes differed from one another by 0.08–0.12%. Our results are also consistent with those of Weir et al. [19] who included two mitochondrial markers. In this case levels of sequence variation for cytochrome oxidase 1 (*cox1*) and NADH dehydrogenase subunit 1 (*nad1*) were the same in both studies; 0.00% and 0.25% variable sequence positions, respectively. Taken together, these comparisons suggest that the sequence diversity observed in the present study is likely to be representative of sequence diversity in *P. agathidicida* more generally.

Given that *P. agathidicida* isolates appear to differ little at the nucleotide sequence level, marker selection must be an important consideration for future evolutionary studies. Work that has documented levels of sequence variation for mitochondrial loci in *Phytophthora* [48] and identified potential marker loci (e.g., [48, 62, 63]) should inform mitochondrial marker selection for *P. agathidicida*. Given the results of Weir et al. [19] similar consideration will need to be given to selection of nuclear markers.

## Divergence time estimates for *P. agathidicida*

Molecular divergence time estimates for stem nodes are often used to test hypotheses about the history of biological lineages. However, we cannot yet confidently draw biogeographical inferences from stem age estimates for *P. agathidicida* because species diversity within *Phytophthora* clade 5 is poorly characterised. We instead investigated the age of the MRCA for a geographically diverse sample of *P. agathidicida* accessions using a combination of the *P. agathidicida* sample collection ages and clock rate estimates from *P. infestans* to calibrate our divergence time analyses. Despite applying information about *P. infestans* clock rates in two different ways we recovered very similar age estimates for the MRCA of the sampled *P. agathidicida*. Specifically, mean estimates fell within two years of each other (i.e., 303.0 and 304.4) and the 95% HPDs broadly overlapped (i.e., 210.1–405.2 and 206.9–414.6).

One question when using secondary rate calibrations is whether the rates applied are appropriate. Tree model adequacy [55] provides a means of evaluating the appropriateness of the tree model underpinning an analysis. Although this approach evaluates the overall tree model, for the current analyses we explicitly compared the observed and simulated results on the basis of three branch length parameters. For both the sequential and simultaneous analyses, all three parameters suggested that observed and simulated results were very similar. This implies that the underlying models could have generated the observed results and therefore that using clock rate estimates from *P. infestans* as priors on divergence time analyses for *P. agathidicida* was appropriate.

Other questions relate to the application of rate calibrations. One is whether the rate priors capture the initial rate estimates. To examine this we compared the probability distributions of the priors and corresponding posteriors from initial and sequential analyses of *P. infestans*. In general these distributions were very similar, although the posterior distributions for the sequential analysis were narrower than those of the priors (S1 Fig). Consistent with the similarity of clock rates, age estimates were also broadly similar for *P. infestans*; for example, crown age estimates for the HERBI lineage [26] were all 252.0–267.0 years (95% HPD, 208.7–342.5 years) old. These results suggest that our priors captured the *P. infestans* clock rates and the associated estimation uncertainty. Another question is whether the priors dominate posteriors. To examine this we compared the distributions of the priors and corresponding posteriors from the sequential analysis of *P. agathidicida* (S2 Fig); in all cases the distributions differ. Furthermore, for both *P. agathidicida* and *P. infestans* the clock rate posteriors from sequential analyses differ from those jointly estimated in the simultaneous analysis (S1 and S2 Figs). Taken together these observations suggest that signal from the *P. agathidicida* sequence data has contributed to age estimates for this species.

## Interpreting the age of the MRCA

If we assume that the inoculum of *P. agathidicida* introduced into New Zealand was genetically uniform, that is it contained a single mitochondrial genome sequence type, then estimates of 303.0–304.4 years (95% HPDs 206.9–414.6 years) old for the MRCA of the sampled *P. agathidicida* isolates are inconsistent with either a recent incursion (i.e., post-1945) or an arrival associated with large scale European settlement (i.e., 1845-1940s). To be consistent with these hypotheses the estimated age of the MRCA would need to have been less than 70 years old or 70–175 years old, respectively. Instead, both mean estimates and 95% HPDs suggest that the MRCA of the sampled isolates is more than 175 years old. Assuming that a genetically uniform inoculum our age estimates suggest *P. agathidicida* was present in New Zealand prior to 1845. Importantly, given the small sample size we expect our analyses to have underestimated the crown age of *P. agathidicida* and for our interpretation to be conservative. Increased sampling

is likely to capture a larger proportion of the underlying genetic diversity and therefore provide older estimates for the crown age of *P. agathidicida*. That is, increased sampling is unlikely to result in estimates suggesting anything other than an arrival prior to 1845.

Our interpretation of divergence time estimates for the MRCA of the sampled *P. agathidicida* isolates assumes that a genetically uniform inoculum arrived in New Zealand. If, instead, multiple mitochondrial genotypes had been introduced then hypotheses suggesting more recent arrival are plausible. Without further analyses we cannot completely exclude the possibility that multiple mitotypes were introduced. That said, multiple introductions do not easily explain the observed geographic distribution of genetic diversity in *P. agathidicida*. If this pathogen had arrived in New Zealand post-1945 then explaining the observed mitogenome diversity would require 9–14 (i.e., 66–100%) of the mitotypes identified in the present analyses to have been introduced. Although multiple introductions are a possibility, in this case we would also need to explain how mutually exclusive subgroups of between two and five sequence types were then moved to the four different geographical regions. Human-mediated transport has been suggested as explaining the spread of *P. agathidicida* (e.g., [24, 64]). However, given a post-1945 introduction as well as the number and geographical distribution of mitotypes our results are difficult to convincingly explain in this way. That human-mediated transport is not clearly linked to the distribution of *P. agathidicida* diversity is consistent with Beachman [27] who, based on forestry records, found little support for widespread human-mediated transport of *P. agathidicida*.

There are several ways in which our inference that *P. agathidicida* arrived in New Zealand prior to widespread European settlement might be further examined. One would be to more fully evaluate the geographic distribution of genetic diversity in *P. agathidicida*. Preliminary analyses of mitochondrial genomes from several additional accessions (e.g., [65]) are consistent with the geographical distribution observed in the present analyses (i.e., the new sequences fall sister to mitotypes from the same location in a phylogenetic network) but further sampling is still needed. Analyses of whole genome data will also be important in this context. For example, based on genome wide analyses of single nucleotide polymorphisms Guo *et al.* [61] identified the same broad geographic groupings. Again a much larger sample will be needed to more fully evaluate the distribution of genetic diversity. Another approach would be to investigate the broader relationships within *Phytophthora* clade 5. There are several reports of diebacks involving other members of the Araucariaceae (e.g., [66]) as well as of clade 5 representatives from elsewhere in the Indo-Pacific (e.g., [67, 68]). Although these examples are obvious candidates for further investigation, a much broader effort (e.g., inclusion of widely sampled isolates [69]) will be needed in order to ensure sampling is sufficient for robust historical inference.

Taken together the available genetic evidence is most consistent with diversification of *P. agathidicida* in New Zealand having begun at least 300 years ago and therefore that this pathogen must have arrived even earlier. Although considerably older than has previously been suggested, this should not be interpreted as evidence of ancient heritage. Indeed, preliminary analyses suggest that *P. agathidicida* and its currently recognised sister species, *P. cocois*, diverged 7,115.9 years (95% HPD, 5514.8–8845.4 years) ago. Further work is clearly needed, but the present analyses appear to be consistent with *P. agathidicida* having been present in New Zealand for between several hundred and several thousand years.

## Implications for management of disease and pathogen risk

Beever et al. [9] recommended that *P. agathidicida* should be managed as a recently introduced pathogen until the origins of this species could be resolved. Further work on origins has not

been conducted; instead it has simply been assumed that *P. agathidicida* arrived in New Zealand post-1945 (e.g., [70, 71]). Several observations (e.g., high host mortality [17]) are consistent with a recent arrival in New Zealand. However, none of these observations provide direct evidence of recent arrival and, significantly, most are also consistent with a longer history in New Zealand.

The disease triangle [72] views disease expression as an emergent property of interactions between a virulent pathogen, a susceptible host, and suitable environmental conditions. Under this model the recent emergence and spread of kauri dieback need not be attributed to recent pathogen arrival. Instead, the pathogen could have remained a relatively benign member of the soil community until changes in the pathogen, host, or environment resulted in conditions conducive to disease expression. Evolutionary changes that increase pathogen virulence or host susceptibility offer a possible explanation. If so, given the discontinuous distribution of disease symptoms, geographic distribution of pathogen diversity, and broad age distribution of infected hosts such evolutionary changes must have occurred on multiple occasions. Although possible, the plausibility of evolutionary explanations is reduced because these changes must have occurred frequently since kauri dieback was first reported in the 1970s but rarely, if at all, before this. In contrast, changes to biotic or abiotic environments offer a more plausible explanation for the emergence of kauri dieback. Studies suggest that even highly aggressive *Phytophthora* species may remain benign unless conditions are conducive to disease [3, 73–76] and, more generally, it is recognized that pathogen activity and disease severity can be moderated by climate (e.g., [77–78]). A recent study [79] has suggested that biotic (e.g., secondary fungal infections) and abiotic (e.g., extremes in rainfall due to climate change) factors have contributed to *Phytophthora*-linked decline of European beech (*Fagus sylvatica* L.) stands in Lower Austria. Human activity has already severely fragmented kauri forests [8, 14, 15] and this could have exacerbated the impacts of other factors such as climate change (e.g., [79]) or potentially co-acting pathogens (e.g., [80]).

It is generally assumed that the distribution of *P. agathidicida* corresponds closely to that of kauri dieback. Yet, if *P. agathidicida* has been long present and disease expression is environmentally mediated then this pathogen may be more widely distributed than is suggested by disease symptoms alone. Pathogen distribution has yet to be assessed in detail [64] but the assumption that the distributions of pathogen and disease closely correspond could potentially compromise research and management efforts. We need to distinguish between "pathogen risk," the risk that *P. agathidicida* has arrived and is persisting at a given location, and "disease risk," the risk that kauri dieback symptoms will be expressed. Pathogen risk is likely to be determined by interactions between the biological characteristics of the pathogen (e.g., life history, dispersal) and landscape or environmental factors that influence movement and survival (e.g., hydrology). Although pathogen risk contributes to disease risk the two should not be considered equivalent. Beyond pathogen presence, disease risk is likely to be mediated by a range of site-specific environmental factors (e.g., microclimate, plant and microbial associations, disturbance history) acting on both pathogen and host. Recognising the contrast between pathogen and disease risk may lead to important insights. For example, potentially providing a means of selecting appropriate management tools and evaluating their effectiveness.

## Supporting information

**S1 Table. Accessions included in the BLAST library used to identify contigs with similarity to oomycete mitochondrial genome sequences.**
(DOCX)

**S1 Fig. Probability distributions of clock rates for *P. infestans*.**
(TIF)

**S2 Fig. Probability distributions of clock rates for *P. agathidicida*.**
(TIF)

**S1 Data. Mitochondrial genome sequences for the 21 accessions of *P. agathidicida* and related clade 5 taxa.**
(FASTA)

## Acknowledgments

We acknowledge the mana whenua of the areas from which the isolates included in this study were collected. SEB acknowledges current and former colleagues on IUFRO Working Party 7.02.09: We thank the International Collection of Microorganisms from Plants (ICMP) culture collection for access to samples and Remco Bouckaert (Auckland University) for help with an earlier version of the divergence time analyses.

## Author Contributions

**Conceptualization:** Richard C. Winkworth, Peter J. Lockhart.

**Formal analysis:** Richard C. Winkworth.

**Investigation:** Richard C. Winkworth, Stanley E. Bellgard, Patricia A. McLenachan.

**Methodology:** Richard C. Winkworth.

**Project administration:** Richard C. Winkworth.

**Resources:** Stanley E. Bellgard.

**Validation:** Richard C. Winkworth.

**Visualization:** Richard C. Winkworth.

**Writing – original draft:** Richard C. Winkworth.

**Writing – review & editing:** Richard C. Winkworth, Stanley E. Bellgard, Patricia A. McLenachan, Peter J. Lockhart.

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
