## [Decision Letter · Decision Letter 0]

26 Jan 2021

PONE-D-20-40042

The mitogenome of *Phytophthora agathidicida*: evidence for a not so recent arrival of the “kauri killing” *Phytophthora* in New Zealand

PLOS ONE

Dear Dr. Winkworth,

Thank you for submitting your manuscript to PLOS ONE. After careful consideration, we feel that it has merit but does not fully meet PLOS ONE’s publication criteria as it currently stands. Therefore, we invite you to submit a revised version of the manuscript that addresses the points raised during the review process.

We look forward to receiving your revised manuscript.

Kind regards,

Jaime E. Blair, PhD

Academic Editor

PLOS ONE

Additional Editor Comments:

Thank you for your submission. The three reviewers and I agree that this manuscript requires minor revision prior to acceptance. Please review all reviewer comments - I have also attached a pdf with my own comments.

3. Please ensure that you refer to Figure 1 in your text as, if accepted, production will need this reference to link the reader to the figure.

4. We note that Figure 1 in your submission contains map images which may be copyrighted. All PLOS content is published under the Creative Commons Attribution License (CC BY 4.0), which means that the manuscript, images, and Supporting Information files will be freely available online, and any third party is permitted to access, download, copy, distribute, and use these materials in any way, even commercially, with proper attribution. For these reasons, we cannot publish previously copyrighted maps or satellite images created using proprietary data, such as Google software (Google Maps, Street View, and Earth). For more information, see our copyright guidelines: http://journals.plos.org/plosone/s/licenses-and-copyright.

(1) You may seek permission from the original copyright holder of Figure(s) [#] to publish the content specifically under the CC BY 4.0 license. 

Reviewers' comments:

Reviewer's Responses to Questions

**Comments to the Author**

1. Is the manuscript technically sound, and do the data support the conclusions?

Reviewer #1: Yes

Reviewer #2: Yes

Reviewer #3: Yes

2. Has the statistical analysis been performed appropriately and rigorously? 

Reviewer #1: Yes

Reviewer #2: Yes

Reviewer #3: Yes

3. Have the authors made all data underlying the findings in their manuscript fully available?

Reviewer #1: Yes

Reviewer #2: Yes

Reviewer #3: No

4. Is the manuscript presented in an intelligible fashion and written in standard English?

Reviewer #1: Yes

Reviewer #2: Yes

Reviewer #3: Yes

5. Review Comments to the Author

Reviewer #1: Winkworth et al. The mitogenome of Phytophthora agathidicida: evidence for a not so recent arrival of the "kauri killing Phytophthora in New Zealand.

This is a nicely manuscript that uses the mitochondrial genomes of 16 P. agathidicida collected from various locations around New Zealand to document the genome diversity and determine divergence times. The authors use genomes of P. infestans to built calibration data for their divergence times estimate of P agathidicida. They conclude that divergence across the genomes was 300+/- years and that P. agathidicida was likely present much longer than was previously reported.

I only have a few concerns or comments to add about the manuscript. Overall, the authors appear to have adequately generated data, analyzed the data and conclude from their results.

However, I question whether comparison to P. infestans for the backbone of the divergence estimates is biologically suitable. P. infestans has a much larger genome and has been shown different when compared to other Phytophthora genomes. P. infestans has undergone a duplication event, which hasn't been observed in other Phytophthora species. It would to see if the divergence estimates of P. agathidicida would be the same if another Phythphthora were used as the backbone of these analyses. Justification for using P. infestans should be included in the manuscript.

Also, it would be good to include an example in the introduction in paragraph 5. Have these analyses been compared before in another fungal system?

I think Table 2 could be summarized or included as a supplemental. Can divergence estimates to one of the other figures?

Reviewer #2: I enjoyed reading this manuscript, it is very interesting and offers a new way forward for management of this pathogen. ie if we assume it is not a recent arrival then we need to deal with the changing environment probably a mix of a changing climate and human activities. instead of being a single disease it becomes a complex disease which generates so many more hypothesis but also more hope. I have made all my comments on the manuscript (highlighted). i have no major comments expect

1. the first part of the results was a bit confusing to me and could probably be more carefully crafted

2. there is a whole paragraph of the discussion that are actually results

Reviewer #3: Winkworth et al. determine the age of the ancestral node of a collection of Phytophthora agathidicida isolates to test the hypothesis of a recent introduction to New Zealand. This is a valuable study, as explained by the authors in the discussion, because it defines needed research on this destructive pathogen as well as management strategies. The authors use mitochondrial genome sequences to conduct the analysis with estimates of mutation rates from P. infestans. Their approach seems justified and reasonable, and the authors use model checking techniques to test the robustness of the result. I do not have any major concerns, only some relatively minor comments for the authors to consider. One small annoyance for future consideration was the lack of page and line numbers in the manuscript.

Introduction:

It is stated that P. infestans has a single host, which is not accurate (potato, tomato, and some other Solanaceous species). It is limited to hosts within single plant family.

Methods:

In first paragraph, “briefly” is misspelled.

The title of Table S1 seems off. These are the species used for mitochondrial genome comparison or assembly, not for LAMP.

Results:

I would like some indication of the sequencing coverage for each genome. Were all genomes able to be closed?

To describe the variation observed, “varied positions” is used and they are described in terms of percentage of the sequence length. These are typically referred to as “segregating sites” in population genetics and molecular evolution. It would be helpful to use other commonly recognized statistics for describing genetic variation, such as nucleotide diversity.

I suggest removing Table 2 as the information in this table could be summarized in a sentence or two in the text, because all the number are essentially identical (it’s not very interesting). The GenBank accession numbers could be added to Table 1. Summarizing the genetic variation observed among P. agathidicida could be more interesting. Also, in the discussion, there is description of genetic variation by geographic region, which should be moved to the results, and might well be presented in a table.

The manuscript is missing the HPPD tree from the BEAST2 analysis. I would be interested to see the divergence times of the geographically defined clades as well.

Discussion:

This sentence is unnecessary: “Specifically, levels of variation at the cytochrome oxidase 1 (cox1) and NADH dehydrogenase subunit 1 (nad1) loci were the same in both cases; 0.00% and 0.25% varied sequence positions, respectively. These comparisons suggest that levels of sequence variation differ between mitochondrial loci.” There is certainly more informative data locus variation in P. agathidicida that the authors could provide, and Frank Martin has made extensive efforts to describe genetic variation in mitochondrial genomes and identify informative regions for markers that is not cited.

I suggest “A further concern is whether clock rate priors dominate the corresponding posteriors.” Does BEAST2 allow analyses to be run without data, with priors only? This is a good way to check if posterior distributions are being determined by priors. The priors could also be added to Figure 4 to better make this point.

6. PLOS authors have the option to publish the peer review history of their article (what does this mean?). If published, this will include your full peer review and any attached files.

Reviewer #1: No

Reviewer #2: No

Reviewer #3: No

---

## [Author Response · Author response to Decision Letter 0]

15 Mar 2021

We have updated the MS in line with the style templates.

In the original submission GenBank accession numbers were not yet available for all the genome sequences. Accession numbers have now been issued for all our sequences and these are included in Table 1.

We have also updated our Data Availability Statement with a full set of GenBank accession numbers.

3. Please ensure that you refer to Figure 1 in your text as, if accepted, production will need this reference to link the reader to the figure.

The figure is now referenced on page 6 (ln 210).

4. We note that Figure 1 in your submission contains map images which may be copyrighted. All PLOS content is published under the Creative Commons Attribution License (CC BY 4.0), which means that the manuscript, images, and Supporting Information files will be freely available online, and any third party is permitted to access, download, copy, distribute, and use these materials in any way, even commercially, with proper attribution. For these reasons, we cannot publish previously copyrighted maps or satellite images created using proprietary data, such as Google software (Google Maps, Street View, and Earth). For more information, see our copyright guidelines: http://journals.plos.org/plosone/s/licenses-and-copyright.

(1) You may seek permission from the original copyright holder of Figure(s) [#] to publish the content specifically under the CC BY 4.0 license.

We have obtained a license from the original copyright holder that is compatible with the requirements of the CC BY 4.0 license used by PLoS ONE. A copy of the copyright permission form is included as requested.

5. Review Comments to the Author

Reviewer #1: Winkworth et al. The mitogenome of Phytophthora agathidicida: evidence for a not so recent arrival of the "kauri killing Phytophthora in New Zealand.

This is a nicely manuscript that uses the mitochondrial genomes of 16 P. agathidicida collected from various locations around New Zealand to document the genome diversity and determine divergence times. The authors use genomes of P. infestans to built calibration data for their divergence times estimate of P agathidicida. They conclude that divergence across the genomes was 300+/- years and that P. agathidicida was likely present much longer than was previously reported.

I only have a few concerns or comments to add about the manuscript. Overall, the authors appear to have adequately generated data, analyzed the data and conclude from their results.

However, I question whether comparison to P. infestans for the backbone of the divergence estimates is biologically suitable. P. infestans has a much larger genome and has been shown different when compared to other Phytophthora genomes. P. infestans has undergone a duplication event, which hasn't been observed in other Phytophthora species. It would to see if the divergence estimates of P. agathidicida would be the same if another Phythphthora were used as the backbone of these analyses. Justification for using P. infestans should be included in the manuscript.

The original manuscript did indicate that using rate estimates from even closely related taxa for temporal calibration has limitations (pg 20, ln 502-503). However, we did not elaborate upon these.

One concern for the reviewer is that the sizes of the P. infestans and P. agathadicida genomes differ. In responding to this it is important to make the distinction between nuclear and mitochondrial genomes.

As the reviewer points out the P. infestans nuclear genome differs from most of the other reported Phytophthora genomes. At ~240 Mb the P. infestans nuclear genome is considerably larger than most of the other reported nuclear genomes; that said there is a broad range of size estimates for Phytophthora species with some similar in size to P. infestans. For example,

P. kernoviae 37-40 Mb

P. cinnamomi 54-78 Mb

P. parasitica 82 Mb

P. sojae 95 Mb

P. cactorum 122 Mb

P. palmivora 135 Mb

P. megakarya 222Mb

P. cambivora 231 Mb

P. alni 236 Mb

Comparative studies suggest that the larger genome size in P. infestans is not related to whole genome duplication (WGD) but has been driven by transposable element (TE) activity (Judelson, 2012; van Hooff et al. 2014). Evidence includes a correlation between TE number and genome size across oomycetes generally as well as P. infestans having fewer gene annotations than other, smaller, Phytophthora genomes (not expected if WGD was responsible for the larger size) (Judelson, 2012). That said, WGD has recently been proposed for P. megakarya, an observation seemingly consistent with increased chromosome size in this species (Morales-Cruz et al. 2020).

We acknowledge that Phytophthora species differ markedly in the evolutionary trends shaping their nuclear genomes. However, our analyses were based on mitochondrial genome sequences that, in contrast to nuclear genomes, are typically more similar to one another. 

Despite exhibiting some structural variation, mitochondrial genome size and gene content appear to be relatively stable (Yuan et al. 2017). Moreover, as part of ongoing work we have compared average non-synonymous changes (dN) for the mitochondrial genes of P. agathidicida and P. infestans (within a data set that included some 50 species) with the values reported by Yuan et al. (2017) (see their Figure 6D). For each of the 34 genes included in our analyses the average dN values for P. agathidicida and P. infestans fall with the inter-quartile ranges reported by these authors. Moreover, there are no consistent differences between values for P. infestans and P. agathidicida (e.g., dN for P. infestans is not always larger than that of P. agathidicida). This latter observation suggests that microevolutionary processes are similar for the P. agathidicida and P. infestans mitochondrial genomes.

At present, rate estimates for the P. infestans mitochondrial genome are the only ones available for Phytophthora. We are unaware of other cases where similar analyses have been attempted or are possible (e.g., where a suitable collect of herbarium samples are available). A comment has been added to indicate this more clearly (pg 11, ln 185-188).

As indicated in the original manuscript we also used TMA to evaluate the appropriateness of the models applied during our analyses. The results of this testing suggests that key features of the data could have been generated by the model used and therefore that using clock priors based on rate estimates for P. infestans to calibrate our analyses is appropriate (pg 21, ln 417-425).

Although we recognise the potential limitations of using secondary rate calibrations, we think that the combination of wider genome level comparisons and testing of our results (e.g., TMA) justifies our use of clock rate estimates from P. infestans to calibrate divergence time analyses in P. agathidicida.

Also, it would be good to include an example in the introduction in paragraph 5. Have these analyses been compared before in another fungal system?

We are not entirely clear what the reviewer is asking for an example of. There are examples from fungi/oomycetes where stem and crown node ages have been reported as part of broad surveys of divergence times as well as those where divergence times have been used to investigate biogeographic history.

We have modified this section of the text and cited examples of the latter (pg 4, ln 90). We are happy to make additional changes if this does not address the reviewers concern.

I think Table 2 could be summarized or included as a supplemental.

This table has now been summarized in the text (pg 14, ln 255-259) with the GenBank accession numbers moved to Table 1 (pg 7-8).

Can divergence estimates to one of the other figures?

Fig 4 has been changed. The chronogram with divergence time estimates has been included and the comparison of evolutionary rates moved to supplementary material (S1 Fig & S2 Fig).

Reviewer #2: I enjoyed reading this manuscript, it is very interesting and offers a new way forward for management of this pathogen. ie if we assume it is not a recent arrival then we need to deal with the changing environment probably a mix of a changing climate and human activities. instead of being a single disease it becomes a complex disease which generates so many more hypothesis but also more hope. I have made all my comments on the manuscript (highlighted). i have no major comments expect

1. the first part of the results was a bit confusing to me and could probably be more carefully crafted

We have extensively revised the first section of the Results (“Mitochondrial genomes”; pg 14-15, ln 246-284). We hope that this revision has improved communication of these points.

2. there is a whole paragraph of the discussion that are actually results

We have moved this paragraph as suggested. The overall description of sequence variation is now in the first section of the Results entitled “Mitochondrial genomes” (pg 15, ln 271-284) with the description of the geographical distribution of this variation incorporated into “Phylogenetic and network analyses” (pg 17, ln 301-312).

Reviewer #3: Winkworth et al. determine the age of the ancestral node of a collection of Phytophthora agathidicida isolates to test the hypothesis of a recent introduction to New Zealand. This is a valuable study, as explained by the authors in the discussion, because it defines needed research on this destructive pathogen as well as management strategies. The authors use mitochondrial genome sequences to conduct the analysis with estimates of mutation rates from P. infestans. Their approach seems justified and reasonable, and the authors use model checking techniques to test the robustness of the result. I do not have any major concerns, only some relatively minor comments for the authors to consider. One small annoyance for future consideration was the lack of page and line numbers in the manuscript.

Line and page numbers have been added.

Introduction:

It is stated that P. infestans has a single host, which is not accurate (potato, tomato, and some other Solanaceous species). It is limited to hosts within single plant family.

This has been corrected (pg 3, ln 47-48)

Methods:

In first paragraph, “briefly” is misspelled.

This has been corrected (pg 6, ln 131).

The title of Table S1 seems off. These are the species used for mitochondrial genome comparison or assembly, not for LAMP.

This has been corrected. The title is now “S1 Table. Accessions included in the BLAST library used to identify contigs with similarity to oomycete mitochondrial genome sequences.”

Results:

I would like some indication of the sequencing coverage for each genome. Were all genomes able to be closed?

The average coverage for each genome has been added to Table 1 (pg 7-8). Yes, all the recovered genomes were closed. In the previous version we had mentioned that complete mitochondrial genome sequences were recovered (ln 240) but we hope this point is made more clearly (pg 14, ln 247-250).

To describe the variation observed, “varied positions” is used and they are described in terms of percentage of the sequence length. These are typically referred to as “segregating sites” in population genetics and molecular evolution. It would be helpful to use other commonly recognized statistics for describing genetic variation, such as nucleotide diversity.

We have replaced “varied positions” with “variable positions” as suggested by the Ed.

I suggest removing Table 2 as the information in this table could be summarized in a sentence or two in the text, because all the number are essentially identical (it’s not very interesting). The GenBank accession numbers could be added to Table 1. 

We have removed Table 2 and moved the GenBank accession numbers to Table 1 (pg 7-8) as suggested. 

The general genome statistics are now briefly described in the second paragraph of the Results (pg 14, ln 252-259).

Summarizing the genetic variation observed among P. agathidicida could be more interesting. Also, in the discussion, there is description of genetic variation by geographic region, which should be moved to the results, and might well be presented in a table.

We have summarized genetic variation within P. agathidicida and moved the description of genetic variation by geographic region to the Results as requested (pg 16, ln 301-312).

The manuscript is missing the HPPD tree from the BEAST2 analysis. I would be interested to see the divergence times of the geographically defined clades as well.

Fig 4 has been changed. The chronogram with divergence time estimates has been included and the comparison of evolutionary rates moved to supplementary material (S1 Fig & S2 Fig). The divergence times of the geographically defined groups are now mentioned in the text (pg 17, ln 336-339; pg 18, ln 361-363).

Discussion:

This sentence is unnecessary: “Specifically, levels of variation at the cytochrome oxidase 1 (cox1) and NADH dehydrogenase subunit 1 (nad1) loci were the same in both cases; 0.00% and 0.25% varied sequence positions, respectively. These comparisons suggest that levels of sequence variation differ between mitochondrial loci.” There is certainly more informative data locus variation in P. agathidicida that the authors could provide, and Frank Martin has made extensive efforts to describe genetic variation in mitochondrial genomes and identify informative regions for markers that is not cited.

In commenting on cox1 and nad1 our purpose was not to suggest that these were the most informative markers. Instead a comparison was being made between our data and that of Weir et al. (2015) who included only cox1 and nad1 in their analyses. The observation being that with similar sample sizes both studies recover similar levels of diversity at these two loci. This, together with the results of Guo et al., suggests that our data, and therefore the resulting age estimates, were representative.

We agree with the reviewer. Studies by Martin and others have identified mitochondrial loci that we would expect to exhibit greater sequence variation than either cox1 or nad1. Indeed our point was that cox1 and nad1 are not the most appropriate mitochondrial loci for marker-based analyses and therefore that future studies should consider other loci.

We recognise that this portion of the discussion may not have been as clear as it needed to have been. We have revised this and now make specific mention of Martin’s work (pg 20, ln 399-404).

I suggest “A further concern is whether clock rate priors dominate the corresponding posteriors.” Does BEAST2 allow analyses to be run without data, with priors only? This is a good way to check if posterior distributions are being determined by priors. The priors could also be added to Figure 4 to better make this point.

Fig 4 has been replaced. The content of the original Fig 4 has been moved to a pair of supplementary files each updated to include the prior distribution as suggested by the reviewer. The text description has also been updated to more fully describe the implications of these results (pg 21, ln 417-440).

---

## [Decision Letter · Decision Letter 1]

7 Apr 2021

The mitogenome of *Phytophthora agathidicida*: evidence for a not so recent arrival of the “kauri killing” *Phytophthora* in New Zealand

PONE-D-20-40042R1

Dear Dr. Winkworth,

We’re pleased to inform you that your manuscript has been judged scientifically suitable for publication and will be formally accepted for publication once it meets all outstanding technical requirements.

Kind regards,

Jaime E. Blair, PhD

Academic Editor

PLOS ONE

Additional Editor Comments (optional):

Thank you for your very careful revision. All reviewers agree that previous comments have been addressed and the manuscript is now acceptable for publication.

Reviewers' comments:

Reviewer's Responses to Questions

**Comments to the Author**

1. If the authors have adequately addressed your comments raised in a previous round of review and you feel that this manuscript is now acceptable for publication, you may indicate that here to bypass the “Comments to the Author” section, enter your conflict of interest statement in the “Confidential to Editor” section, and submit your "Accept" recommendation.

Reviewer #1: All comments have been addressed

Reviewer #2: All comments have been addressed

Reviewer #3: All comments have been addressed

2. Is the manuscript technically sound, and do the data support the conclusions?

Reviewer #1: (No Response)

Reviewer #2: Yes

Reviewer #3: (No Response)

3. Has the statistical analysis been performed appropriately and rigorously? 

Reviewer #1: (No Response)

Reviewer #2: Yes

Reviewer #3: (No Response)

4. Have the authors made all data underlying the findings in their manuscript fully available?

Reviewer #1: (No Response)

Reviewer #2: Yes

Reviewer #3: (No Response)

5. Is the manuscript presented in an intelligible fashion and written in standard English?

Reviewer #1: (No Response)

Reviewer #2: Yes

Reviewer #3: (No Response)

6. Review Comments to the Author

Reviewer #1: (No Response)

Reviewer #2: Thank you for your very careful review of the manuscript, all my comments have been addressed and I think the manuscript will instigate some very interesting debate in New Zealand re the management of the disease.

Reviewer #3: (No Response)

7. PLOS authors have the option to publish the peer review history of their article (what does this mean?). If published, this will include your full peer review and any attached files.

Reviewer #1: No

Reviewer #2: No

Reviewer #3: No

---

## [Editor Report · Acceptance letter]

14 May 2021

PONE-D-20-40042R1 

The mitogenome of *Phytophthora agathidicida*: evidence for a not so recent arrival of the “kauri killing” *Phytophthora* in New Zealand 

Dear Dr. Winkworth:

I'm pleased to inform you that your manuscript has been deemed suitable for publication in PLOS ONE. Congratulations! Your manuscript is now with our production department. 

Kind regards, 

on behalf of

Dr. Jaime E. Blair 

Academic Editor

PLOS ONE